# Risk of colorectal cancer in patients with diabetes mellitus: A Swedish nationwide cohort study

Uzair Ali Khan[1,2], Mahdi Fallah[1,3]ʘ*, Kristina Sundquist[3,4,5], Jan Sundquist[3,4,5], Hermann Brenner[1,6,7], Elham Kharazmi[1,3,8]ʘ

**1** Division of Preventive Oncology, German Cancer Research Center (DKFZ) and National Center for Tumor Diseases (NCT), Heidelberg, Germany, **2** Medical Faculty Heidelberg, Heidelberg University, Heidelberg, Germany, **3** Center for Primary Health Care Research, Lund University, Malmö, Sweden, **4** Department of Family Medicine and Community Health, Department of Population Health Science and Policy, Icahn School of Medicine at Mount Sinai, New York, New York, United States of America, **5** Center for Community-based Healthcare Research and Education (CoHRE), Department of Functional Pathology, School of Medicine, Shimane University, Izumo, Japan, **6** Division of Clinical Epidemiology and Aging Research, German Cancer Research Center (DKFZ), Heidelberg, Germany, **7** German Cancer Consortium (DKTK), German Cancer Research Center (DKFZ), Heidelberg, Germany, **8** Statistical Genetics Group, Institute of Medical Biometry and Informatics, Heidelberg University, Heidelberg, Germany

ʘ These authors contributed equally to this work.
* m.fallah@dkfz.de (MF)

## Abstract

### Background

Colorectal cancer (CRC) incidence is increasing among young adults below screening age, despite the effectiveness of screening in older populations. Individuals with diabetes mellitus are at increased risk of early-onset CRC. We aimed to determine how many years earlier than the general population patients with diabetes with/without family history of CRC reach the threshold risk at which CRC screening is recommended to the general population.

### Methods and findings

A nationwide cohort study (follow-up:1964–2015) involving all Swedish residents born after 1931 and their parents was carried out using record linkage of Swedish Population Register, Cancer Registry, National Patient Register, and Multi-Generation Register. Of 12,614,256 individuals who were followed between 1964 and 2015 (51% men; age range at baseline 0–107 years), 162,226 developed CRC, and 559,375 developed diabetes. Age-specific 10-year cumulative risk curves were used to draw conclusions about how many years earlier patients with diabetes reach the 10-year cumulative risks of CRC in 50-year-old men and women (most common age of first screening), which were 0.44% and 0.41%, respectively. Diabetic patients attained the screening level of CRC risk earlier than the general Swedish population. Men with diabetes reached 0.44% risk at age 45 (5 years earlier than the recommended age of screening). In women with diabetes, the risk advancement was 4 years. Risk was more pronounced for those with additional family history of CRC (12–21 years

**Data Availability Statement:** This study made use of the Swedish Cancer Registry as well as the Swedish National Patient Register. Data from these registers cannot be shared by study authors,

however further information and relevant contact details can be found on: https://www.socialstyrelsen.se/en/statistics-and-data/registers/register-information/ Links and email addresses for registers: https://www.socialstyrelsen.se/en/statistics-and-data/registers/register-information/swedish-cancer-register/ (cancerregistret@socialstyrelsen.se) https://www.socialstyrelsen.se/en/statistics-and-data/registers/register-information/the-national-patient-register/ (patientregistret@socialstyrelsen.se).

**Funding:** UAK was funded by the Helmholtz Association for German Research Centers (https://www.helmholtz.de/en/). The funder had no role in study design, data collection and analysis, decision to publish, or preparation of the manuscript.

**Competing interests:** The authors have declared that no competing interests exist.

**Abbreviations:** COPD, chronic obstructive pulmonary disease; CRC, colorectal cancer; DM, diabetes mellitus; FDR, first-degree relative; HNPCC, hereditary nonpolyposis colorectal cancer; IBD, inflammatory bowel disease; ICD, International Classification of Diseases.

earlier depending on sex and benchmark starting age of screening). The study limitations include lack of detailed information on diabetes type, lifestyle factors, and colonoscopy data.

## Conclusions

Using high-quality registers, this study is, to our knowledge, the first one that provides novel evidence-based information for risk-adapted starting ages of CRC screening for patients with diabetes, who are at higher risk of early-onset CRC than the general population.

## Author summary

### Why was this study done?

- Diabetes is associated with increased risk of colorectal cancer (CRC), especially in young adults before age 50.
- CRC incidence is increasing among young adults who are not targeted for screening.
- Diabetes has not been considered as a risk factor in any CRC screening guideline.

### What did the researchers do and find?

- For each single age, we calculated the risk of developing CRC in the next 10 years; for example, at age 50, which is the most common age for starting CRC screening, the risk of developing CRC during next 10 years (age 50 to 59) in the Swedish population was 0.44% in men and 0.41% in women.
- Men and women with diabetes reached the risk levels for 50-year-old individuals (0.44% and 0.41%, respectively) at about age 45 instead of age 50, i.e., nearly 5 years earlier than the general population, whereas patients with an additional family history of CRC reach these screening risk thresholds, 12 to 21 years earlier than the general population.

### What do these findings mean?

- These findings for the first time provide evidence-based information about the best starting age of screening for CRC in patients with diabetes.
- A major strength of this study would relate to the extremely large and comprehensive national (Swedish) datasets available and the duration involved (all Swedish residents born after 1931 and their parents, followed up to 2015).
- Clinicians could inform patients with diabetes (with or without family history of CRC) about this possibility and encourage individualized counseling for CRC screening.

## Introduction

Colorectal cancer (CRC) has become the third most common cancer worldwide and is second in cause of death due to cancer, despite being a preventable disease [1]. Since the emergence of CRC screening, myriad studies have demonstrated that screening for CRC is more beneficial than for any other major malignancy and that screening is more cost-effective than not screening [2,3]. In the United States, it has been highlighted that since the introduction of CRC screening, CRC incidence rates have declined [4]. However, it was found that the trend for all ages hid patterns that existed in young people. Since the 1980s, incidence in those aged 20 to 39 increased 1.0% to 2.4% per year, for those aged 40 to 54 incidence increased 0.5% to 1.3% annually and markedly, an adult born in 1990 was observed to have twice the risk of CRC at the same age as an adult born in 1950 [4]. Similar patterns are observed in Europe, where an investigation of 143 million young adults across 20 countries showed CRC incidence rapidly rising in those who are below age 50 years [5]. This trend has been observed rather globally among young individuals, and therefore, screening guidelines should be adjusted accordingly [5,6].

Few efforts have been put forth to combat the issue of rising CRC incidence in young adults. Strategies have included lowering the age of screening for all individuals regardless of risk, which has high financial burdens [7]. Alternatively, it has been suggested to identify risk factors that make young individuals particularly high risk and personalize the screening procedure. However, as of yet, most countries recommend a single average-risk age for CRC screening (most commonly at age 50) [8]. Few risk factors have been highlighted for earlier CRC screening in guidelines that are centered around family history, inflammatory bowel disease (IBD), or rare genetic disorders, which alone cannot account for the widespread increase in young-onset CRC worldwide [8,9]. Hence, it is believed that targeting high-risk young people specifically is the best and least invasive approach, and investigating risk factors in young people is the way to combat this issue [3].

Diabetes mellitus and CRC share common risk factors and are both becoming more prevalent in young adults, and diabetes diagnosis has been consistently associated with an increased risk of CRC later in life. A recent study has also shown that having a diabetes diagnosis before the age of 50 increases the risk of early-onset CRC nearly 2-fold [10]. Despite this, diabetes has never been indicated in CRC screening guidelines as a risk factor [8]. Identifying potential risk groups for early-onset CRC has clinical significance if high-risk individuals are made aware of their risk and potentially screened earlier. We aimed to determine whether individuals with a diabetes diagnosis with and without family history of CRC reach the CRC risk of their peers in the general population at younger ages, and if so, how many years earlier. We used high-quality data from several long-standing nationwide Swedish registers, which resulted in, to the best of our knowledge, the world's largest and most robust study of its kind.

## Methods

In this study, we used data from several nationwide registers from Sweden for all individuals born in Sweden since 1931 and their parents. The study dataset was created through the linkage of the data from Multi-Generation Register, Death Register, Swedish Cancer Registry, and national censuses using unique lifetime registration numbers assigned to all residents. The Multi-Generation Register contains genealogical information. The Death Register provides information on date of death, and the national censuses offer data on participants' migration records and similar demographic measures. The linked Swedish Cancer Registry data carried information on date of cancer diagnosis, tumor topography and morphology, and detailed diagnostic reports from physicians for the period 1958 to 2015. All cancer records were

reported using International Classification of Diseases (ICD) codes from versions 7 through 10. In the linked Swedish family–cancer datasets, there were about 13 million individuals with genealogical information, of which more than 160,000 were patients with CRC diagnosed during the cancer registry period 1964 to 2015.

The abovementioned datasets, the Swedish National Patient Registers, which include data from all private and public hospitals and specialized doctor visits in Sweden, were linked together using pseudonymized identification numbers (S1 Fig). Hospital (inpatient) records from 1964 to 2015 and day clinic records from 2001 to 2015 with detailed information on disease diagnosis and date of visit were available for this study. Information on patients with diabetes was extracted using ICD codes (ICD-7: 260; ICD-8: 250; ICD-9: 250; ICD-10: E10, E11, E13, and E14). All individuals with pregnancy- and malnutrition-related diabetes as well as those with a diabetes diagnosis following a CRC diagnosis were excluded. We recognized family history of CRC in first-degree relatives (FDRs). The study follow-up for all individuals in the analysis was defined as: starting from the most recent of birth year, immigration year, or 1964; ending at the earliest of CRC diagnosis date, emigration year, death year, or 2015. The final dataset contained maximum 51 years of follow-up from beginning of 1964 to end of 2015. A flowchart of the final study population is presented in the Supporting information (S2 Fig).

In the analysis, personal history of diabetes and family history of CRC were treated as time-dependent variables. This means that all individuals were only recorded as diabetic cases from the year in which they were diagnosed. Similarly, an individual was only recorded as having CRC family history from the year in which the FDR was diagnosed. The rationale behind utilizing the dynamic (time dependent) method is that it is understood to be the most appropriate for studies involving risk stratification since it provides real-time risk estimates that can be applied in clinical settings [11]. For instance, if a nondiabetic individual wants to know their risk of developing CRC at the present time, only their known histories can be taken account even if they were to become diabetic later in life. Alternatively, the static (traditional method of ascertaining family and personal disease history in studies) method is possible in register-based studies where an individual's entire prior personal or family histories are known at the conclusion of study follow-up. Resultantly, the static method would be most appropriate for estimating the effect size that a certain risk factor has on an outcome. We chose to employ the dynamic method since our primary aim was to provide risk-adapted starting age of CRC screening in patients with diabetes that could be used for real-time counseling. Furthermore, the dynamic method reflects the time-varying nature of disease histories, making it ideal for the purposes of this study.

The main outcome measure in the analysis was 10-year cumulative risk, i.e., the risk (%) of developing CRC within the next 10 years at each age. Risk-adapted screening ages in patients with diabetes were calculated using 10-year cumulative risk of CRC. The 10-year cumulative risks were calculated using the following formulas:

- *Age-specific incidence rate = Total cases at each age (every 1 year) divided by the total person-years at that age*

- *10-year cumulative rate for age X = Sum of 10 consecutive yearly age-specific incidence rates from age X to age X+9*

- *10-year cumulative risk = 1 − exp$^{(-10\text{-year cumulative rate})}$*

Rather than aggregating cumulative incidence by age groups (the traditional method of calculating cumulative risk), age-specific precise values from individual participant's yearly data were used in the calculation [12]. Comparing 10-year cumulative risk in each risk group to the population 10-year cumulative risk allowed the inference of risk-adapted

screening ages. A smoothing effect to reduce random variation in incidence rates was employed using a moving average. For instance, for the 10-year cumulative risk at age 30, the average of the 10-year cumulative risks at ages 29, 30, and 31 was used, while for age 31, the average of the 10-year cumulative risks at ages 30, 31, and 32 was used, and so on. This method of calculating risk-adapted starting age of cancers has already been used for some other conditions [13–15].

Using this method, we could provide the age at which patients with diabetes with/without family history of CRC reached a similar level of CRC risk to that of a 50-year-old individual in the general population, the most commonly recommended age of first screening in guidelines [8]. We also provided the same for 45, 55, and 60 year olds as they represent the variability in starting ages of CRC screening globally. Covariates included age and sex. As a sensitivity analysis, we repeated the 10-year cumulative risk analysis in men, removing all individuals with a prior diagnosis with IBD, an established CRC risk factor, to ensure they did not confound our analysis [16]. All statistical analyses were conducted using SAS statistical program version 9.4 (by SAS Institute, Cary, North Carolina, USA). To avoid risk of identification of participants, researchers had only access to pseudonymized secondary data. The main analyses were planned before starting the execution of data analyses. However, further analyses have been performed to answer reviewers' comments, such as adding supplementary tables of basic characteristics and a table for 10-year cumulative risk by age group, with no influence on our main findings. No data-driven changes to analyses took place.

### Ethics statement

The study protocol was approved by the Lund Regional Ethics Committee (2012/795).

## Results

From the beginning of follow-up, a total of 12,614,256 individuals with genealogical information were included in the analysis (51% men; age range at baseline 0 to 107 years). From this population, 162,226 patients with CRC were identified. Additionally, a total of 559,375 patients with diabetes were identified, and 101,135 (18%) of them were diagnosed before age 50. Among patients with diabetes, the mean time to CRC diagnosis was 5.8 years. Further characteristics of patients with diabetes and patients with CRC are presented in **Tables 1** and **2**. The 10-year cumulative risk estimates of developing CRC in patients with diabetes with and without family history of CRC by sex and age group are presented as the Supporting information in **S1 Table**.

### Benchmark age 50

Our results in terms of 10-year cumulative risk (**Figs 1** and **2**) showed that for 50-year-old men in the general Swedish population, risk of developing CRC within the next 10 years was 0.44%. The 10-year cumulative risk for 50-year-old women in the general Swedish population was 0.41%. Men with no family history of CRC but with a diabetes diagnosis before age 50 reached the same 10-year cumulative risk of CRC as 50-year-old men in the general Swedish population at age 45, i.e., 5 years earlier, whereas women with no family history of CRC but with a diabetes diagnosis before age 50 were observed to reach the same 10-year cumulative risk as 50-year-old women in the general Swedish population at age 46, i.e., 4 years earlier. Men and women with diabetes and family history of CRC attained the population level of 10-year cumulative risk at age 32 (18 years earlier) and age 38 (12 years earlier), respectively. Men without diabetes or a CRC family history reached the population level of risk at age 51 (1 year later).

**Table 1. Characteristics of patients with diabetes in study population.**

| | Patients with diabetes | | | | | |
| | All | | Without CRC | | With CRC | |
| | *N* | % | *N* | % | *N* | % |
|---|---|---|---|---|---|---|
| **Total** | 559,375 | 100.0 | 547,839 | 97.9 | 11,536 | 2.1 |
| **Sex** | | | | | | |
| Men | 288,348 | 51.5 | 281,609 | 51.4 | 6,739 | 58.4 |
| Women | 271,027 | 48.5 | 265,230 | 48.4 | 4,797 | 41.6 |
| **Age at DM diagnosis** | | | | | | |
| <20 | 28,639 | 5.1 | 28,601 | 5.22 | 38 | 0.3 |
| 20–29 | 15,196 | 2.7 | 15,121 | 2.76 | 75 | 0.7 |
| 30–39 | 20,373 | 3.6 | 20,198 | 3.69 | 175 | 1.5 |
| 40–49 | 37,066 | 6.6 | 36,549 | 6.67 | 517 | 4.5 |
| 50–59 | 76,678 | 13.7 | 75,107 | 13.7 | 1,571 | 13.6 |
| 60–69 | 130,909 | 23.4 | 127,239 | 23.2 | 3,670 | 31.8 |
| 70–79 | 153,043 | 27.4 | 148,863 | 27.2 | 4,180 | 36.2 |
| 80–84 | 97,471 | 17.4 | 96,161 | 17.6 | 1,310 | 11.4 |
| **Period of diagnosis** | | | | | | |
| 1964–1969 | 5,466 | 1.0 | 5,406 | 1.0 | 60 | 0.5 |
| 1970–1979 | 57,752 | 10.3 | 56,646 | 10.3 | 1,106 | 9.6 |
| 1980–1989 | 114,024 | 20.4 | 111,702 | 20.4 | 2,322 | 20.1 |
| 1990–1999 | 137,054 | 24.5 | 133,876 | 24.4 | 3,178 | 27.5 |
| 2000–2009 | 147,111 | 26.3 | 143,800 | 26.2 | 3,311 | 28.7 |
| 2010–2015 | 97,968 | 17.5 | 96,409 | 17.6 | 1,559 | 13.5 |
| **Disease history** | | | | | | |
| IBD | 19,232 | 3.4 | 18,848 | 3.4 | 384 | 3.3 |
| HNPCC | 82 | 0.0 | 74 | 0.0 | 8 | 0.1 |
| Obesity* | 19,019 | 3.4 | 18,705 | 3.4 | 314 | 2.7 |
| Alcohol use disorder* | 20,074 | 3.6 | 19,733 | 3.6 | 341 | 3.0 |
| COPD* | 52,096 | 9.3 | 50,970 | 9.3 | 1,126 | 9.8 |

COPD, chronic obstructive pulmonary disease; CRC, colorectal cancer; DM, diabetes mellitus; HNPCC, hereditary nonpolyposis colorectal cancer; IBD, inflammatory bowel disease; *N*, number of people; %, percentage of patients with diabetes with the specified characteristic out of total number of patients with diabetes.

*Hospitalization or visit to specialty outpatient clinics for these conditions.

## Benchmark age 45

Our results in terms of 10-year cumulative risk (**Table 3**) showed that for both 45-year-old men and women in the general Swedish population, risk of developing CRC within the next 10 years was 0.24%. Men with no family history of CRC but with a diabetes diagnosis before age 45 reached the same 10-year cumulative risk of CRC as 45-year-old men in the general Swedish population at age 40, i.e., 5 years earlier, whereas women with no family history of CRC but with a diabetes diagnosis before age 45 reached the same risk level as 45-year-old women in the general Swedish population at age 42, i.e., 3 years earlier. Men and women with diabetes and family history of CRC attained the population level risk at age 31 (14 years earlier).

## Other benchmark ages (55 and 60)

As different countries have different benchmark ages for initiation of CRC mass screening in the population (ranging from 45 in the US to 55 to 60 in the United Kingdom), we

**Table 2. Characteristics of patients with CRC in study population.**

| | Patients with CRC | | | | | |
| | All | | Nonfamilial CRC | | Familial CRC | |
| | N | % | N | % | N | % |
|---|---|---|---|---|---|---|
| **Total** | 162,226 | 100.0 | 155,247 | 95.7 | 6,979 | 4.3 |
| **Sex** | | | | | | |
| Men | 85,212 | 52.5 | 81,245 | 52.3 | 3,808 | 54.6 |
| Women | 77,014 | 47.5 | 74,002 | 47.7 | 3,171 | 45.4 |
| **Age at diagnosis** | | | | | | |
| <20 | 428 | 0.3 | 427 | 0.3 | 1 | 0.0 |
| 20–29 | 920 | 0.6 | 897 | 0.6 | 23 | 0.3 |
| 30–39 | 2,347 | 1.4 | 2,221 | 1.4 | 126 | 1.8 |
| 40–49 | 7,160 | 4.4 | 6,676 | 4.3 | 484 | 6.9 |
| 50–59 | 20,238 | 12.5 | 18,840 | 12.1 | 1,398 | 20.0 |
| 60–69 | 42,534 | 26.2 | 40,019 | 25.8 | 2,515 | 36.0 |
| 70–79 | 53,577 | 33.0 | 51,646 | 33.3 | 1,931 | 27.7 |
| ≥80 | 35,022 | 21.6 | 34,521 | 22.2 | 501 | 7.2 |
| **Period of diagnosis** | | | | | | |
| 1964–1969 | 5,400 | 3.3 | 5,398 | 3.5 | 2 | 0.0 |
| 1970–1979 | 15,901 | 9.8 | 15,880 | 10.2 | 21 | 0.3 |
| 1980–1989 | 26,141 | 16.1 | 25,686 | 16.5 | 455 | 6.5 |
| 1990–1999 | 36,236 | 22.3 | 35,617 | 22.9 | 619 | 8.9 |
| 2000–2009 | 45,586 | 28.1 | 42,942 | 27.7 | 2,644 | 37.9 |
| 2010–2015 | 32,962 | 20.3 | 29,724 | 19.1 | 3,238 | 46.4 |
| **Age at diabetes diagnosis** | | | | | | |
| <50 | 805 | 0.5 | 738 | 0.5 | 67 | 1.0 |
| ≥50 | 10,731 | 6.6 | 10,252 | 6.6 | 479 | 6.9 |
| All ages | 11,536 | 7.1 | 10,990 | 7.1 | 546 | 7.8 |
| **Disease history** | | | | | | |
| IBD | 6,198 | 3.8 | 5,662 | 3.6 | 536 | 7.7 |
| HNPCC | 103 | 0.1 | 0 | 0.0 | 103 | 1.5 |
| Obesity* | 2,918 | 1.8 | 2,747 | 1.8 | 171 | 2.5 |
| Alcohol use disorder* | 4,660 | 2.9 | 4,456 | 2.9 | 204 | 2.9 |
| COPD* | 13,324 | 8.2 | 12,618 | 8.1 | 706 | 10.0 |

COPD, chronic obstructive pulmonary disease; CRC, colorectal cancer; DM, diabetes mellitus; HNPCC, hereditary nonpolyposis colorectal cancer; IBD, inflammatory bowel disease; N, number of people; %, percentage of patients with specified characteristic out of total number of patients with CRC.

*Hospitalization or visit to specialty outpatient clinics for these conditions.

provided risk-adapted starting ages of CRC screening for different benchmark ages (45, 50, 55, and 60 years; **Table 3**). Those with a personal history of diabetes and no family history of CRC reached the population level of risk 4 to 5 years earlier than the general Swedish population for benchmark ages of screening 55 and 60. By contrast, those with both diabetes and family history of CRC reached the general Swedish population risk 21 years earlier (men) and 14 to 15 years earlier (women). Finally, both men and women without diabetes and CRC family history reached the population level of risk 1 year later than the general Swedish population (age 56 for benchmark age 55 and age 61 for benchmark age 60).

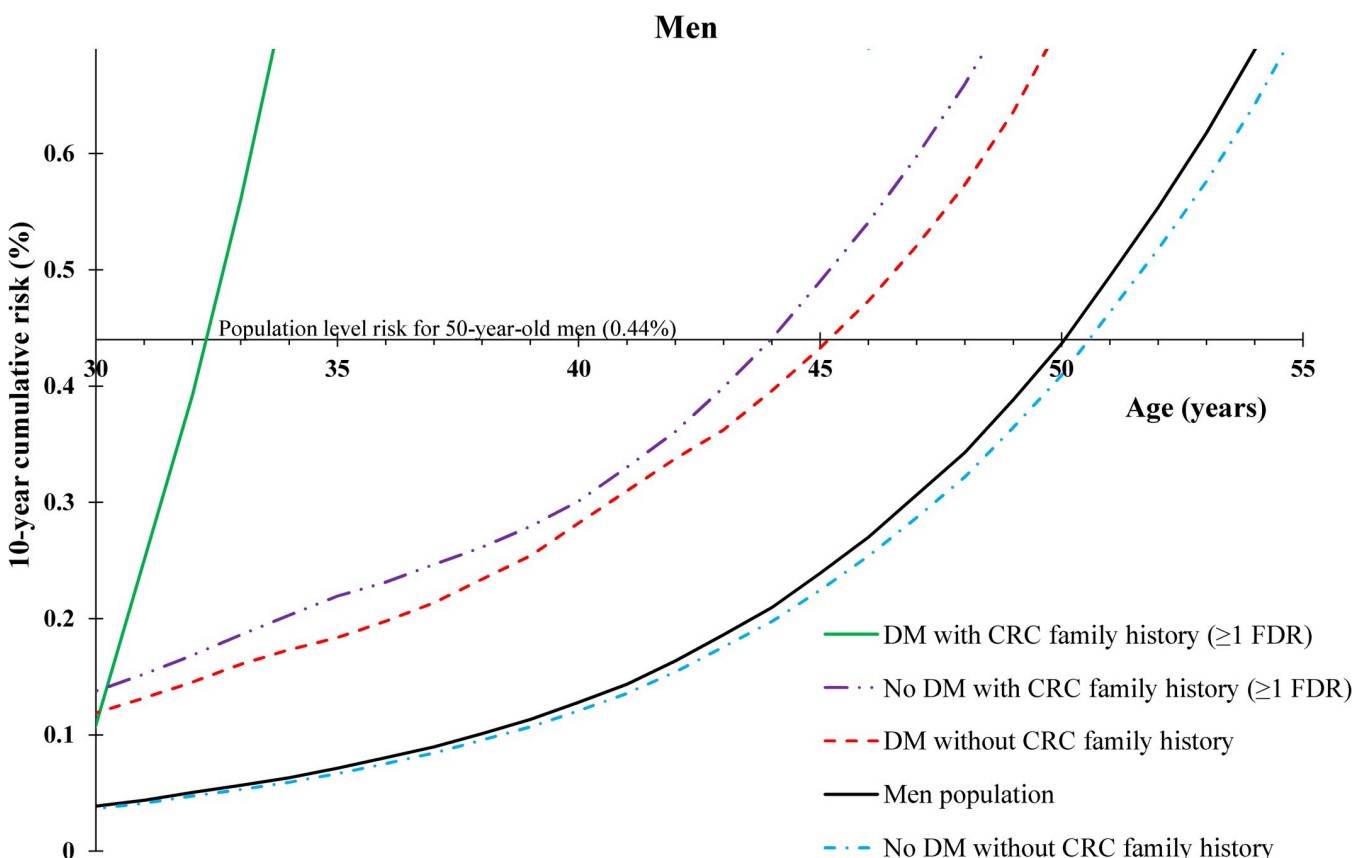

**Fig 1. Age-specific 10-year cumulative risk of CRC by personal history of DM before age 50 and family history of CRC in FDRs among <u>men.</u>** CRC; colorectal cancer; DM, diabetes mellitus; FDR, first-degree relative.

## Comparison with existing guidelines

A comparison between our findings for patients with diabetes with a CRC family history and established screening guidelines for individuals with an FDR with CRC revealed a wide range of difference between our recommended risk-adapted starting ages of screening and those in the current guidelines (from 5 to 21 years), although the difference for other example ages could be even higher (**S2 Table**). Such a difference for patients with diabetes without family history of CRC was 3 to 5 years depending on sex and benchmark ages of mass screening (**Table 3**).

## Ten-year cumulative risk after removing patients with IBD

We also excluded patients with IBD from our analysis and did not find any changes of substance to our results. A total of 445,444 cases of IBD (185,869 men; 44%) were excluded from the analysis. Of all IBD cases, 5,957 (1,613 men; 27%) preceded a CRC diagnosis, and 19,232 IBD cases (6751 men; 35%) were comorbid with diabetes. No substantial changes in our main estimates were detected after exclusion of IBD cases.

## Discussion

Using several high-quality Swedish nationwide registers, we found that patients with diabetes, without family history of CRC, reach the same level of CRC risk as 50-year-old individuals in

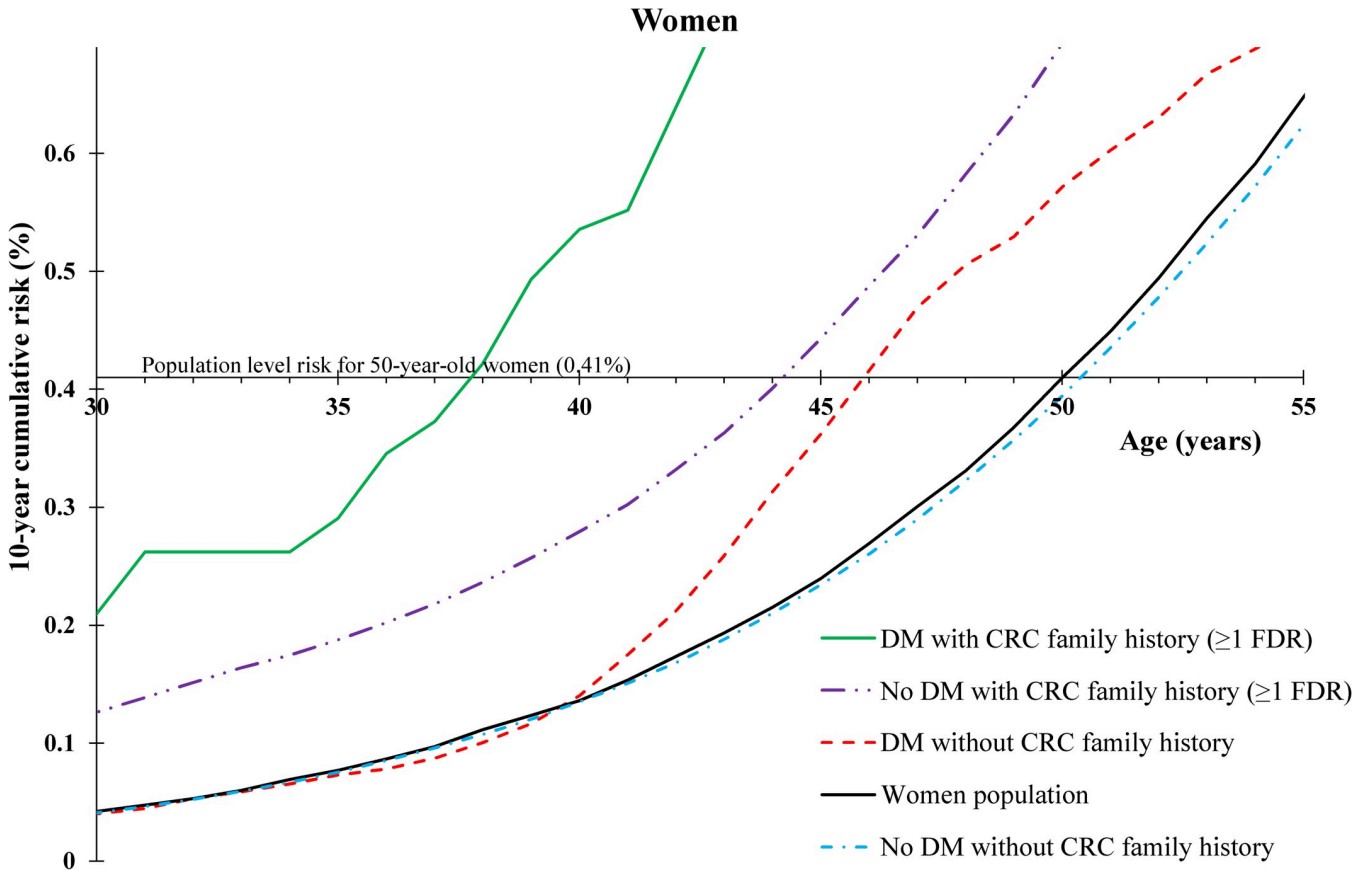

**Fig 2. Age-specific 10-year cumulative risk of CRC by personal history of DM before age 50 and family history of CRC in FDRs among <u>women.</u>** CRC, colorectal cancer; DM, diabetes mellitus; FDR, first-degree relative.

the general Swedish population 4 to 5 years earlier. This risk advancement in patients with diabetes with family history of CRC was 18 years earlier for men and 12 years earlier for women compared to their peers in the general Swedish population. Depending on the benchmark age of mass screening in the general population and sex, patients with both diabetes and family history of CRC attained the population level risk 12 to 21 years earlier.

The associations between diabetes, family history of CRC, and CRC risk have been already reported [17–19]. However, there has been no study to date that assessed how these risk associations can be used in clinical counseling of patients with diabetes with and without family history of CRC and offered risk-adapted starting ages of CRC screening for them. Our current study provided this novel and clinically useful information. Another novel aspect of this study in comparison to others that investigated CRC risk in patients with diabetes is the use of 10-year cumulative risk to plot changes in CRC risk by age [18].

In our study, we compared 10-year cumulative of CRC risk for different combinations of sex, age, CRC family history, diabetes status, and benchmark ages for starting screening. We used a benchmark age of 50 years as an example since this is the recommended age of first screening by most CRC screening guidelines [8]. Our results show that patients with diabetes reach the Swedish population level of 10-year cumulative risk several years earlier, but when also considering that young patients with diabetes have a much higher risk of early-onset CRC as opposed to late-onset CRC, screening even in the 30s might be warranted in people with both CRC family history and diabetes. Although CRC screening in the 30s is unusual, when

**Table 3. Risk-adapted starting ages of CRC screening by sex, personal history of DM and family history of CRC tailored to different benchmark stating age of mass screening in the population.**

| Sex | Diabetes personal history[†] | CRC family history | Patients (Obs) | Risk-adapted starting age of screening (years) | | | |
|---|---|---|---|---|---|---|---|
| Population[*] | Any | Any | 162,226 | **45** | **50** | **55** | **60** |
| Men | No | No | 75,120 | 45 | 51 | 56 | 61 |
| | Yes | No | 6,388 | 40 | 45‡ | 50 | 55 |
| | Yes | ≥1 FDR | 351 | 31 | 32 | 34 | 39 |
| Women | No | No | 69,137 | 45 | 50 | 56 | 61 |
| | Yes | No | 4,602 | 42 | 46 | 51 | 55 |
| | Yes | ≥1 FDR | 195 | 31 | 38 | 41 | 45 |

CRC, colorectal cancer; CRC patients (Obs), cumulative number of observations with CRC within the groups; bold ages indicate benchmark starting ages of CRC screening in the general Swedish population; DM, diabetes mellitus; FDR, first-degree relative.

[*]Ten-year cumulative risks of CRC in the general Swedish population at ages 45, 50, 55, and 60 were 0.24%, 0.44%, 0.77%, and 1.28% in men, and 0.24%, 0.41%, 0.65%, and 0.98% in women, respectively.

[†]DM was diagnosed before CRC diagnosis and benchmark starting age of mass screening in the population, i.e., diabetes diagnosis age <45 for benchmark screening age 45, diabetes diagnosis age <50 for benchmark screening age 50, etc.

‡Example: 45-year-old men with a personal history of DM without family history of CRC reached the same 10-year cumulative risk of CRC as 50-year-old men in the general population who were subject to CRC screening in their society, i.e., with a benchmark starting age of mass screening in the general population at age 50 years, the risk-adapted starting age for those with only personal history of DM was 45 years; thus, those with a personal history of DM without family history of CRC could be screened at age 45 years, 5 years earlier than the general population.

considering that the mean time for an adenomatous polyp to progress to CRC is between 10 and 12 years [20] and that CRC incidence is rapidly rising in those below screening age, it could be justified. Although the results of randomized trials of colonoscopy use are yet to be learned, elevated CRC rates in young adults have been observed and need action [21–23]. It has also been reported that overall CRC screening is effective and cost-effective and that a risk-adapted approach is the best [2,24]. Our findings showed that risk-adapted CRC screening by diabetes personal history with and without family history of CRC might be beneficial. Furthermore, similar trends in 10-year cumulative risk of CRC in both men and women with diabetes demonstrate internal validity of our results, and minor differences are in line with known higher risk of CRC in men than in women. It is noteworthy, however, that the evaluation of cost-effectiveness of risk-adapted CRC screening, specifically for patients with diabetes, warrants further investigation.

Our study benefited from several high-quality Swedish nationwide register datasets, including Swedish Cancer Registry, Multi-Generation (genealogy) Register, national censuses, and Inpatient and Outpatient Registers with roughly half a century of follow-up. These resources enabled us to design the world's largest and most robust study of its kind. All datasets were linked through pseudonymized identification number, removing traditional limitations of studies, such as biases due to self-reporting CRC diagnosis, family history of CRC, and also diagnosis of diabetes. Furthermore, this long-term cohort study allowed us to establish CRC incidence over time with 10-year cumulative risk so as to measure risk dynamically with age. This is a more detailed look at CRC risk as compared to just the use of relative risk measures, such as standardized incidence ratio or hazard ratio, used by most population-based studies since we were able to compare all risk groups at various ages, rather than produce a single estimate of relative risk [17]. Another strength of this study was the use of time-dependent history of diseases. Since we had precise information on date of diagnosis of CRC in individuals, in their family members, and date of diabetes diagnosis, we were able to ensure all instances of CRC family history and diabetes diagnosis occurred before CRC diagnosis. This means that

we were able to avoid potential issues of reverse causation. The time-dependent method in this study is preferred for risk stratification and identifying individuals for risk-adapted screening since it reflects the dynamic nature of developing diabetes and diagnosis of CRC in family members [11]. In addition, we were able to avoid limitations common in most studies that treat disease history as static conditions, such as immortal time bias and exposure misclassification by ensuring individuals were considered as diabetic cases from the date of diagnosis and non-cases until that point.

One of the limitations of our study was minimal access to data on lifestyle factors. Type 2 diabetes and CRC share several risk factors including obesity and lack of regular physical activity [25–27]. However, previous cohort studies have shown that controlling for common risk factors of CRC and type 2 diabetes, such as obesity and diet, does not significantly modify CRC risk estimates in patients with diabetes [28,29]. In a related study, we had data on hospitalization for chronic obstructive pulmonary disease (COPD, a surrogate measure for smoking), obesity, and alcohol use disorder. Adjustment for these risk factors did not alter those results.

We could not stratify our analyses by diabetes type because the ICD codes for diabetes diagnosis in our dataset did not accurately differentiate the type of diabetes until 1997 (ICD-10) and even after that the majority had both diagnoses, which might correspond to older definition of insulin-dependent and non-insulin–dependent diabetes mellitus rather than the actual type of diabetes. Type 1 diabetes (which does not share risk factors with CRC like type 2 diabetes and usually is diagnosed early in life) has also been implicated with a higher risk of CRC. This suggests that the association between diabetes and CRC is not purely dependent on lifestyle factors and therefore, irrespective of type, is an ideal candidate for risk-adapted CRC screening [30].

Another limitation was lack of colonoscopy data to ensure elevated risk of CRC was not confounded by the possibility that patients with diabetes and patients with CRC family history are more likely to be screened for CRC. In a related study, we evaluated risk of CRC in patients with diabetes by calendar period and did not find substantial differences in risk of familial CRC [31]. The lack of CRC screening data also did not have a significant impact on our findings and the potential implication of their application. This is because a nationwide organized CRC screening does not exist in Sweden. An organized screening as an official recommendation (not a law) has been introduced only as a pilot phase in 2008 in the Swedish Stockholm Gotland area merely for age 60 to 69, where even invitational coverage accounted for less than 9% of the nationwide screening-eligible population (age 50 to 74) [32]. Furthermore, patients with diabetes have been recognized to be poor at adhering to diabetes treatment recommendations [33], making it unlikely that they would seek out CRC screening more so than a person in the general population. As a sensitivity analysis, we also removed patients with IBD from our analysis to ensure they did not confound the association between diabetes and CRC and found minimal attenuation to the results.

Since there is a wide disparity in CRC screening guidelines globally for age of first screening, such as age 55 in the Netherlands, age 55 or 60 in the UK (depending on location), age 45 in the US, and age 50 in most other countries such as Germany [8,34], we provided results for various benchmarks. In fact, the applied method can be "personalized" to fit any population or any preferred benchmark age of initial screening in the general population. We found that for all benchmark ages of screening, those with combined CRC family history and diabetes personal history reach the Swedish population level of 10-year cumulative risk much earlier than CRC family history and diabetes personal history individually, suggesting that both criteria contribute to CRC risk differentially. Regardless of the specific benchmark, however, the results of our study may be informative for the development of personal risk calculators, which

possibly in combination with other established factors or in combination with genetic risk scores [35–37], and used for calculating personalized starting ages of screening in the future. The method of integrating such results into other risk prediction models with more risk factors of cancer (but no information on diabetes) have been discussed elsewhere [15]. Further discussions around the importance of earlier screening in patients with diabetes and the generalizability of our findings have been included in **S1 eDiscussion** as a Supplementary information, which also contains explanation about age-specific incidence of CRC in Sweden over time (**S3 Fig**).

## Conclusions

The present study provides population-based evidence for potential risk-adapted starting ages of CRC screening in patients with diabetes with and without family history of CRC. With CRC incidence rising among young adults and the accumulation of evidence associating diabetes with early-onset CRC risk, we observed that patients with diabetes in Sweden reach the general population level of CRC risk several years earlier. Patients with both diabetes and family history of CRC reached the population level of risk 1 to 2 decades earlier than the general Swedish population. Irrespective of disparity and uncertainty regarding the optimal age of screening for average risk individuals globally, our evidence-based results propose a novel risk group who may benefit from earlier initial screening. Despite lack of data regarding type of diabetes and lifestyle factors, our findings warrant investigation into the potential advantages, disadvantages, and efficacy of screening patients with diabetes earlier. Our findings thereby assist to consider a risk-adapted approach to CRC screening or at the very least can be used to inform those with diabetes about how many years earlier than the general population they could initiate CRC screening.

## Supporting information

**S1 Fig. Study dataset: Information from several databases (A, B, C, and D) were compiled together to form the study dataset including about 13 million individuals with valid genealogical and cancer data with follow-up from 1964 to 2015.** (A) Information on family relationship for all individuals Sweden residents born after 1931 and their parents. (B) Information on cancer diagnosis, including year/age of diagnosis and International Classification of Diseases (ICD) codes from 1958 to 2015. (C) Information on diabetes mellitus diagnosis, including year/age of diagnosis, number of hospital (or day clinic) visits and ICD codes from 1964 (or 2000) to 2015. (D) Information on follow-up time, including birth, immigration/emigration, and death.
(TIF)

**S2 Fig. Flowchart of study population.**
(TIF)

**S3 Fig. Age-specific incidence of colorectal cancer in Sweden over time (1962 to 2015) by sex.**
(TIF)

**S1 Table. Sex and age-specific 10-year cumulative risk of colorectal cancer in population and different risk groups by personal history of diabetes (diagnosed before age 50) and family history of colorectal cancer.**
(DOCX)

**S2 Table. Comparison between recommended risk-adapted starting ages of screening in the US, Canada, and UK Guidelines and our evidence-based ones.**
(DOCX)

**S1 eDiscussion. Supporting information.**
(DOCX)

**S1 RECORD Checklist. Supporting information.**
(DOCX)

## Author Contributions

**Conceptualization:** Mahdi Fallah, Elham Kharazmi.

**Formal analysis:** Uzair Ali Khan.

**Investigation:** Uzair Ali Khan, Mahdi Fallah, Elham Kharazmi.

**Methodology:** Mahdi Fallah, Elham Kharazmi.

**Resources:** Mahdi Fallah, Kristina Sundquist, Jan Sundquist.

**Software:** Mahdi Fallah.

**Supervision:** Mahdi Fallah, Elham Kharazmi.

**Visualization:** Elham Kharazmi.

**Writing – original draft:** Uzair Ali Khan, Mahdi Fallah, Elham Kharazmi.

**Writing – review & editing:** Uzair Ali Khan, Mahdi Fallah, Kristina Sundquist, Jan Sundquist, Hermann Brenner, Elham Kharazmi.

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
