## [Editor Report · Decision Letter 0]

6 Jul 2020

Dear Dr Fallah, 

Thank you for submitting your manuscript entitled "Risk-adapted colorectal cancer screening in patients with diabetes mellitus: A nationwide cohort study" for consideration by PLOS Medicine.

Your manuscript has now been evaluated by the PLOS Medicine editorial staff and I am writing to let you know that we would like to send your submission out for external peer review.

Kind regards,

Helen Howard, for Clare Stone PhD 

Acting Editor-in-Chief

PLOS Medicine 

plosmedicine.org

---

## [Decision Letter · Decision Letter 1]

5 Aug 2020

Dear Dr. Fallah,

Thank you very much for submitting your manuscript "Risk-adapted colorectal cancer screening in patients with diabetes mellitus: A nationwide cohort study" (PMEDICINE-D-20-03051R1) for consideration at PLOS Medicine. 

Your paper was evaluated by a senior editor and discussed among all the editors here. It was also evaluated by three independent reviewers, including a statistical reviewer. The reviews are appended at the bottom of this email and any accompanying reviewer attachments can be seen via the link below:

[LINK]

In light of these reviews, I am afraid that we will not be able to accept the manuscript for publication in the journal in its current form, but we would like to consider a revised version that addresses the reviewers' and editors' comments. Obviously we cannot make any decision about publication until we have seen the revised manuscript and your response, and we plan to seek re-review by one or more of the reviewers. 

We expect to receive your revised manuscript by Aug 26 2020 11:59PM. Please email us (plosmedicine@plos.org) if you have any questions or concerns.

We look forward to receiving your revised manuscript. 

Sincerely,

Emma Veitch, PhD

PLOS Medicine

On behalf of Clare Stone, PhD, Acting Chief Editor,

PLOS Medicine

plosmedicine.org

*Please structure your abstract using the PLOS Medicine headings (Background, Methods and Findings, Conclusions - Methods and Findings should be a single subsection). 

*In the last sentence of the Abstract Methods and Findings section, please include a brief note about any key limitation(s) of the study's methodology.

*At this stage, we ask that you include a short, non-technical Author Summary of your research to make findings accessible to a wide audience that includes both scientists and non-scientists. The Author Summary should immediately follow the Abstract in your revised manuscript. This text is subject to editorial change and should be distinct from the scientific abstract. Please see our author guidelines for more information: https://journals.plos.org/plosmedicine/s/revising-your-manuscript#loc-author-summary

*Please clarify in the paper if the analytical approach reported here corresponded to one laid out in a prospective protocol or analysis plan. Please state this (either way) early in the Methods section.

*We'd suggest using an appropriate guideline, such as the TRIPOD guideline (for reporting of prediction models for prognosis or diagnosis) to help ensure full reporting of the methods and findings of this study - https://www.equator-network.org/reporting-guidelines/tripod-statement/. If using the guideline please note this and cite the guideline in your Methods section, and include a completed TRIPOD checklist as supporting information with the revised paper. 

Comments from the reviewers:

Reviewer #1: "Risk-adapted colorectal cancer screening in patients with diabetes mellitus: A nationwide cohort study" employed lifetime data from over 12 million patients linked from various Swedish registers and datasets, to examine the effect of diabetes mellitus and/or family history of colorectal cancer (CRC) on an individual's ten-year cumulative risk of CRC. It was found that diabetes mellitus caused a risk advancement of about five years from the recommended screening age of fifty, compared to reference individuals without diabetes mellitus. A family history of CRC resulted in even more pronouned CRC risk advancements, of 12-21 years. This suggests that risk-adapted starting age of CRC screening may be appropriate, as summarized in Table 1.

A major strength of this study would relate to the extremely large and comprehensive national (Swedish) datasets available, and the duration involved (all Swedish residents born after 1931, followed up to 2015). Some queries however remain, mainly relating to the details of cumulative risk computation, and treatment of confounders:

1. Details on the computation of the main outcome of 10-year cumulative risk, were not found in the Methods section. To the best of our knowledge, there are various plausible ways by which cumulative risk can be estimated, together with accompanying parameters (e.g. smoothing of risk between years/within age groups). These details might be provided (possibly in supplementary data), since it appears central to the findings. If the technique used was also basically the same as relevant prior work, it might also be stated.

A related article [11] by some of the authors was briefly referenced in the discussion as the "time-dependent method in this study", but that appears to have various configurations (i.e. accumulative, static, dynamic). The authors might consider going into similar depth as [11], in describing the actual method applied, without which statistical validity is difficult to further assess.

2. It might be considered to also directly include relevant data (e.g. risk-adapted CRC screening ages by age of FDR diagnosis) instead of citing prior work ([26] in this case), if possible.

3. The utility of diabetes mellitus as a risk factor would appear to depend on the availability of diabetes screening/diagnosis. As such, the authors might briefly discuss the prevalence/frequency/method/other characteristics of diabetes screening/diagnosis for this population, and if possible provide relevant statistics such as the observed prevalence of diabetes by age, and average onset of CRC in years after diabetes was diagnosed.

4. All available plausible confounders (e.g. IBD, COPD?) from the registers might be systematically listed, although it is mentioned as a limitation that there was minimal access to data on lifestyle factors (e.g. obesity, recognized as a major risk factor in [9])

5. A demographics summary table and flowchart of the subject selection procedure might be considered.

6. Minor issue: on Page 10, "did no alter those results" might be "did not".

Reviewer #2: This is a well performed study and the authors should be commended. 

1. It would be better to provide the actual 10-yr cumulative risk for the different categories (with confidence limits) as the first table. Table 1 could then be a separate Table 2 (with perhaps abbreviated foot notes)

2. The challenge with this type of study is to know if there are any confounders with a DM diagnosis - eg. obesity, diet, tobacco use, etc. Although the authors acknowledge this limitation in the discussion, is it possible (even if not available in the full data set) to know what the frequency of co-variates might be, and then provide a discussion around the potential impact of that? 

3. The authors should be commended for doing a sensitivity analysis excluding IBD. Please provide details, including the number of patients excluded, gender, DM diagnosis. 

4. Although screening begins for many at the age of 50, the reality is that the penetrance of screening is very poor. The authors suggest that they did not have access to colonoscopy data. Alternatively, could the authors provide model of 10-year risk prior to the implementation of widespread screening? The rationale for the question is that if the screening rate is high in this population, it may underestimate the impact of earlier screening for patients with DM in a population where the penetrance of screening is low. 

5. The discussion should be shortened. 

Reviewer #3: The authors utilized a national-wide database to provide evidence on the starting age of colorectal cancer screening considering the familial history of colorectal cancer as well as the history of diabetes. The topic is interesting and provides an important implication, however, more details for the methods and results should be provided. 

Major comments

I can not find ref 26, which used the same database and addressed FDR as a determinant of starting age of colorectal cancer screening. It seems that the results of two articles overlap substantially.

Table 1: The results are too simple. Please provide more details such as person-year, cumulative incidence at each age, etc. Why was not the concurrent DM history and family history considered? Did the "patients (Obs)" refer the cumulative numbers of colorectal patients within the groups? Please provide the results for the total population for men and women, respectively. 

Was the proportion of patients with a family history comparable with other studies conducted in Sweden? Please provide more details on the distribution of FDR, e.i. mother/father/siblings. 

Considering the long study period, the authors may consider the period-cohort effects of colorectal cancer incidence. 

Please provide the comparisons between the results of the current study and existing screening recommendation for the person with a family history of colorectal cancer. 

Minor comments

Page 11 the last sentence: I do not understand the intended meaning of the sentence. 

Page 12: The starting age of screening in the UK is not 60.

[LINK]

---

## [Decision Letter · Decision Letter 2]

1 Oct 2020

Dear Dr. Fallah,

Thank you very much for re-submitting your manuscript "Risk-adapted colorectal cancer screening in patients with diabetes mellitus: A nationwide cohort study" (PMEDICINE-D-20-03051R2) for review by PLOS Medicine.

I have discussed the paper with my colleagues and the academic editor and it was also seen again by reviewers. I am pleased to say that provided the remaining editorial and production issues are dealt with we are planning to accept the paper for publication in the journal.

[LINK]

We look forward to receiving the revised manuscript by Oct 08 2020 11:59PM. 

Sincerely,

Adya Misra, PhD

Senior Editor 

PLOS Medicine

plosmedicine.org

Requests from Editors:

Title- please amend to "Risk of colorectal cancer among patients ..." and add Sweden in the title

Short title and throughout please amend “diabetic patients” to patients with Type 2 diabetes

Abstract methods and findings- the first sentence needs some revision, perhaps “was carried out” or similar to end the sentence.

Abstract methods and findings- please provide participant demographics like age range, sex etc

Abstract and throughout- please ensure you specify which type of diabetes you mean. If this is Type 2 or insulin dependent- please mention that. 

Abstract-please provide brief details on the data sources used for this work. For example- which inpatient/outpatient register was used and which cancer dataset?

Abstract methods and findings- please mention the statistical analyses undertaken and the last sentence of this section should explicitly outline 2-3 limitations of your study design/methodology. We suggest you add “The study limitations include … “

Abstract conclusions-please avoid assertions of primacy by adding “to our knowledge” or similar

Abstract conclusions- Please remove the word “personalised” as it is a bit misleading

Data availability- we understand that the nationwide data cannot be shared by study authors. Could you please provide names and links to various data sources used in this study, noting that these cannot be shared by study authors but that interested parties may be able to request access by contacting the relevant persons. Please add contact details for the same. In addition, please note study authors cannot act as sole gatekeepers to datasets. 

Author byline- we do not require author designations here, please remove all iterations of “group leader” “PhD students” etc

Author summary- this sentence requires simplification “Diabetic patients were observed to reach the population level of risk, at which screening starts, about 5 years earlier, and accordingly could start counseling for colorectal cancer screening earlier than the general population”. Please describe the analyses undertaken, in general language for accessibility to a general readership. 

Author summary section “what did the researchers do and find” points 3 and 4 go considerably beyond what you may include in this section as it touches on application of your work. Please remove these points.

Throughout- please use square brackets for references

Page 5, please add to our knowledge before “which resulted in the world’s largest and most robust study of its kind”.

Page 8 first paragraph- please reword “averaged” to “average of” or other suitable alternatives 

Throughout the submission where you say “nationwide” please specify which country. When giving specific numbers of years of earlier disease onset should include "in the Swedish population

Page 15 – please rephrase “ poor lifestyle choices” to be specific and not stigmatising 

Page 15- please replace “alcoholism” with high alcohol intake or similar 

RECORD checklist- please use paragraphs and sections instead of page numbers as these are likely to change 

Please discuss the current level of adherence to CRC screening in the discussion and how this affects your results as well as potential implications for application.

Please move Suppl tables 1 and 2 to the main manuscript

Overall all conclusions need to be tempered, such as "... reached the same level of CRC risk" as this is an observational study. In the same vein, please add limitations of your work, including information on potential confounders. 

Comments from Reviewers:

Reviewer #1: The authors have addressed our previous concerns sufficiently. Figures 1 & 2 might also show the case for No DM but with CRC family history, since that combination appears to be missing.

[LINK]

---

## [Editor Report · Decision Letter 3]

19 Oct 2020

Dear Dr. Fallah, 

On behalf of my colleagues and the academic editor, Dr. Aesun Shin, I am delighted to inform you that your manuscript entitled "Risk of colorectal cancer in patients with diabetes mellitus: A Swedish nationwide cohort study" (PMEDICINE-D-20-03051R3) has been accepted for publication in PLOS Medicine. 

PRODUCTION PROCESS

Before publication you will see the copyedited word document (within 5 business days) and a PDF proof shortly after that. The copyeditor will be in touch shortly before sending you the copyedited Word document. We will make some revisions at copyediting stage to conform to our general style, and for clarification. When you receive this version you should check and revise it very carefully, including figures, tables, references, and supporting information, because corrections at the next stage (proofs) will be strictly limited to (1) errors in author names or affiliations, (2) errors of scientific fact that would cause misunderstandings to readers, and (3) printer's (introduced) errors. Please return the copyedited file within 2 business days in order to ensure timely delivery of the PDF proof. 

If you are likely to be away when either this document or the proof is sent, please ensure we have contact information of a second person, as we will need you to respond quickly at each point. Given the disruptions resulting from the ongoing COVID-19 pandemic, there may be delays in the production process. We apologise in advance for any inconvenience caused and will do our best to minimize impact as far as possible.

PRESS

PROFILE INFORMATION

Thank you again for submitting the manuscript to PLOS Medicine. We look forward to publishing it. 

Best wishes, 

Adya Misra, PhD

Senior Editor 

PLOS Medicine

plosmedicine.org